# Pharmacological Abortion in a Pandemic: An Italian Medico-Legal Perspective

**DOI:** 10.3390/ijerph182212043

**Published:** 2021-11-16

**Authors:** Clara Cestonaro, Anna Aprile, Matteo Bolcato, Daniele Rodriguez, Alessandro Feola, Giulio Di Mizio

**Affiliations:** 1Legal Medicine, University of Padua, 35121 Padua, Italy; clara.cestonaro@hotmail.it (C.C.); anna.aprile@unipd.it (A.A.); danielec.rodriguez@gmail.com (D.R.); 2Department of Experimental Medicine, University of Campania “Luigi Vanvitelli”, Via Luciano Armanni 5, 80138 Naples, Italy; alessandro.feola@unicampania.it; 3Forensic Medicine, Department of Law, “Magna Graecia” University of Catanzaro, 88100 Catanzaro, Italy; giulio.dimizio@unicz.it

**Keywords:** abortion, RU486, mifepristone, pandemic, COVID-19, legal medicine

## Abstract

The limitations caused by the spread of the SARS-CoV2 virus have had repercussions on the voluntary termination of pregnancy. During the pandemic, Italy issued updated guidelines regarding voluntary termination of pregnancy by means of mifepristone and prostaglandin. This included news concerning the time limit and location in which this procedure could be accessed: updates partially recognize women’s needs, and they are into line with the European parliament’s recent exhortations. However, these updates do not change the previously provided responsibilities that lie with doctors. This article aimed to compare regulations concerning medical abortion in Italy and other countries, with a focus on recent Italian updates in the context of pandemic.

## 1. Introduction

An important aspect of public health, in terms of the organization of and access to healthcare services, is the protection of women’s health and autonomy by means of termination of pregnancy. As a result of SARS-CoV2 pandemic-related restrictions [1], which have resulted in reduced access to hospital facilities for non-infection-related healthcare services [2,3], some countries have extended access to pharmacologically induced abortions, whereas others have attempted to impede access to abortions by deeming such procedures as non-essential.

The aim of this article was to compare the current regulations in force in Italy with those in other countries as regards pharmacologically induced termination of pregnancy, with particular reference to updates introduced in the context of the current pandemic.

## 2. Medical Abortion

### 2.1. Mifepristone

‘Medical abortion’ usually indicates the ‘early pregnancy termination (…) performed without primary surgical intervention and resulting from the use of abortion-inducing medications’. Clinically tested procedures for medical abortion emerged during the 1950s, although the idea of terminating a pregnancy with the use of medicines began centuries earlier [4].

Mifepristone, coded RU486, the main drug currently used in this procedure, was developed by Étienne-Émilie Baulieu from the Roussel-Uclaf pharmaceutical company. It was approved by the French government and placed on the market in 1988 with the name Mifegyne [5]. Mifepristone is a synthetic drug that performs its antiprogestin function by competitively binding the progestin receptors, inhibiting the effects of endogenous and exogenous progesterone; in order to induce a medical abortion, this drug is used in combination with the prostaglandin analog misoprostol [6]. The combination of mifepristone and misoprostol may cause complications attributable to both the prostaglandin (such as nausea, vomiting, fever, and flushing) and medical abortion (such as hemorrhage and infections) [7]. An article published in 2006 noted that when inducing an abortion with mifepristone, ‘the rate of failure to cause complete termination of pregnancy increases dramatically along with hemorrhagic events, as the gestational age and the size of the placenta increases’ [8]. In 1992, Berer noted that patients, in reference to medical abortions, felt a greater sense of control and less invasiveness [9]. Some of the reasons identified for terminating a pregnancy medically rather than surgically included the fact that it could be carried out in a more ‘relaxed’ environment compared to an operating room [10].

Mifepristone, initially introduced to terminate pregnancies up to the 49th gestational day, was subsequently considered by international medical societies in their guidelines for use up to the 63rd day [11].

### 2.2. Pharmacologically Induced Termination of Pregnancy: An International Perspective

After France, mifepristone was approved for use in the United Kingdom in 1991 and in Sweden in 1992. According to the principle of mutual recognition, it was registered in 1999 and approved for use in Austria, Belgium, Denmark, Finland, Germany, Greece, Luxembourg, the Netherlands, and Spain [12].

A discussion paper of the British Medical Association (BMA) in 2017 reported that under the Abortion Act in England, Wales, and Scotland, mifepristone and misoprostol must be administered in National Health Service (NHS) or in other approved premises and, having received the second dose, the woman could return home to complete abortion [13]. Indeed, the recommendations issued by the Royal College of Obstetricians and Gynaecologists (RCOG) in 2011 indicated that it was ‘safe and acceptable’ for women to leave the facility after the administration of misoprostol ‘to complete the abortion at home’ and that the existence of an ‘adequate support strategy and follow-up arrangements’ was needed [14].

In France, prior to the COVID-19 pandemic, pharmacologically induced voluntary termination of pregnancy was permitted up to the seventh week from the last menstruation (five weeks of pregnancy) if performed in a ‘cabinet médical’ (doctor’s office) and up to the ninth week (seven weeks of pregnancy) if performed in an ‘établissement de santé’ (health institution) [15].

In the United States, in 2000, the Food and Drug Administration (FDA) approved the use of mifepristone along with prostaglandin for termination of pregnancy up to 49 days from the last menstruation [16]. In line with FDA regulations, mifepristone was not available in pharmacies and was only distributed to doctors who could ‘accurately determine the duration of a patient’s pregnancy and detect an ectopic pregnancy’. The FDA stipulated that prescribing doctors must also be ‘able to perform surgery in cases of an incomplete abortion or severe bleeding’ or that they must have organized in advance for such procedures to be performed by ‘other qualified physicians’ [17]. The regimen approved by the FDA in 2016 included the use of Mifeprex (mifepristone) with misopristol within 70 days of gestation, indicating–in line with the Risk Evaluation and Mitigation Strategy (REMS)–the need for prescription and dispensing ‘by or under the supervision of a healthcare provider’ with the appropriate qualifications, and dispensing in clinics, medical offices, and hospitals ‘by or under the supervision of a certified healthcare provider’ [18].

Mifepristone was approved in 2001 by the health authorities in New Zealand, where it is available at authorized abortion institutions. It may be prescribed by ‘any medical practitioner’ and must be administered ‘by a healthcare professional in a licensed premise’. In 2012, the drug was also registered in Australia. In line with the directions given by the Royal Australian and New Zealand College of Obstetricians and Gynecologists (RANZCOG), the prescriber ‘must supervise and take responsibility’ for the entire abortion procedure, from the administration of mifepristone to the confirmation of the success of the abortion and follow-up. The RANZCOG highlighted, however, that there was a large body of evidence to support self-administration of misoprostol at home within the first 63 days of gestation [19].

In a handbook issued in 2014, the World Health Organization (WHO) suggested that ‘allowing home use of misoprostol following provision of mifepristone at a health care facility can improve the privacy, convenience and acceptability of services, without compromising on safety’ and suggested the use of healthcare facilities for performing procedures in pregnancies that have exceeded 63 days and for managing serious complications. Moreover, the importance of women to ‘access advice and emergency care in the event of complications, if necessary’ was highlighted [20].

In 2018, the WHO indicated that ‘where there is access to a source of accurate information and to a health-care provider (…), the abortion process can be self-managed with pregnancies <12 weeks of gestation without the direct supervision of a health-care provider’ (highlighting there is little evidence for pregnancies above 10 weeks). The report proceeded to list specific healthcare professionals (including obstetricians, nurses, and specialist and non-specialist doctors) recommended ‘for provision of medical abortion of pregnancies <12 weeks’ [21].

### 2.3. Pharmacologically Induced Termination of Pregnancy in Italy

In Italy, the option of pharmacologically induced abortion was met with strong opposition, since it was believed that said option could be perceived as an incentive to resort to the voluntary termination of pregnancy (VTP) [22].

Though mifepristone has been used in various regions of Italy since 2005 [23], the Italian Medicines Agency (*Agenzia Italiana del Farmaco* (AIFA)), via resolution No. 14 [24], approved marketing authorization of Mifegyne only in 2009. The use of this medicine had to ‘be applied in strict adherence with the regulatory precepts provided for by Law No. 194, 22 May 1978 to ensure and protect women’s health’, guaranteeing admission to one of the healthcare facilities provided for by Article 8 of Law No. 194/78 ‘from the moment of drug administration until the expulsion of the product of conception’. The Agency proceeded to indicate that the ‘abortion procedure must take place under supervision of a doctor from the obstetrics-gynecology service’. Article 8 of said law [25] sets forth that the termination of a pregnancy must be ‘carried out by a doctor in the obstetrics-gynecology department in a general hospital (…)’ and that ‘this procedure can be carried out in specialized public hospitals’. Furthermore, within the first 90 days, this procedure may be carried out in ‘care homes authorized by the Region’ provided they meet the requirements and in ‘appropriately equipped public outpatient clinics connected to hospitals and authorized by the Region’.

Resolution No. 1460, dated 24 November 2009 [26], provides, in reference to the therapeutic indications for Mifegyne, for the ‘medical termination of an intrauterine pregnancy (used in sequential association with prostaglandin, the use of which, in Italy, is authorized up to 49 days of amenorrhea)’; ‘the softening and dilatation of the cervix uteri prior to surgical termination of pregnancy in the first trimester’; the ‘preparation for the action of prostaglandin analogues in the therapeutic termination of the pregnancy (beyond the first trimester)’; and ‘inducing labor in the event of intrauterine fetal death in patients where prostaglandin and oxytocin cannot be used’.

On 24 June 2010, the Ministry of Health approved guidelines for the use of mifepristone and prostaglandin in voluntary terminations of pregnancy [27], stating that the *Consiglio Superiore di Sanità* (Council of Health) deemed ‘regular admissions’ necessary for all stages of the ‘abortion procedure’, specifically referring to opinions issued by the Council itself, issued in 2004 (‘risks of pharmacologically induced abortion are to be considered equivalent to the risks of surgical abortion only if the termination of pregnancy takes place in the hospital setting’ [28]) and in 2005 (‘therefore, the combination of mifepristone and misoprostol must be administered in public hospitals or in another facility required by the aforementioned law, and the woman must be held there until the abortion is complete’ [29]).

The aforementioned guidelines stipulated, as part of the treatment admission criteria, the requirement of an ‘intrauterine pregnancy up to 49 days of amenorrhea/ultrasound date of gestation up to 35 days’.

In addition, the Council of Health stated on 18 March 2010 that it deemed it ‘necessary, for the purposes of compliance with Law No. 194/78 throughout the national territory, that the pharmacologically induced termination of pregnancy procedure be performed as a regular admission until the verification of the complete expulsion of the product of conception’, recommending that guidelines be drafted in line with both medical and surgical VTP-related data [30].

It is thus clear that strict time and location restrictions have been required for abortion procedures; Iadicicco [22] said that they have been inspired by a persistent ideological aversion to abortion rather than by the desire to control scientific uncertainties.

Furthermore, in the years immediately following the authorization of mifepristone for commercial use, recourse to this method in Italy was irregular: according to the Ministry of Health’s data, the combination of mifepristone + prostaglandin was not used in Abruzzo and Calabria in 2010, nor in Marche in 2011. In addition, a high percentage of women (76%) decided to self-discharge following the administration of mifepristone or before the expulsion of the product of abortion was complete, subsequently returning to hospital to complete the procedure [23]. In a report published in 2018, the Ministry of Health indicated an increasing trend in recourse to medical abortion between 2014 and 2017 [31].

## 3. Pharmacologically Induced Termination of Pregnancy and the Pandemic

As a result of the SARS-CoV2 pandemic, interventions considered as non-essential have been limited, with repercussions on voluntary terminations of pregnancy as well as on other sectors.

In the United States of America, several states, especially the Southern and Midwestern states and Alaska, have restricted access to abortion procedures, deeming them non-essential [32]. Though certain states have placed restrictions on surgical abortions only, other states have ordered or supported the cessation of both surgical and medical abortions [33,34].

As regards medical abortions, the American College of Obstetricians and Gynecologists (ACOG), in the October 2020 issue of the Practice Bulletin, stated: ‘although the FDA REMS program for mifepristone continues to require dispensing in the clinician’s office, the U.S. labeling for mifepristone no longer indicates that the medication should be used only in the clinician’s office’. The Level A ACOG recommendations contained in that document continued stating that ‘any clinician with the skills to screen patients for eligibility for medication abortion and to provide appropriate follow-up can provide medication abortion’ and that women could safely use both mifepristone and misoprostol at home [35].

As regards the situation in Europe, a study conducted by Moreau et al. (conducted on 46 countries/regions by means of a survey executed by local experts or a desk review conducted by the authors themselves) highlighted significant discrepancies in the levels of access to abortion procedures during the pandemic. The study also showed that for non-medical reasons, ‘abortion care’ has been prohibited in Andorra, Malta, Monaco, San Marino, Liechtenstein, and Poland and suspended in Hungary owing to a ‘ban on non-life-threatening surgeries in state hospitals’. Conversely, some countries have expanded the use of medical abortion at home. In particular, England, Wales, Northern Ireland, Scotland, France, and Finland have ‘officially expanded home medical abortion (…) either as a new service delivery option or through expansion of the gestational age eligibility limit’. Moreau et al. reported in analytical tables, which we have referenced, changes to abortion services in 10 European countries in the context of COVID-19 and abortion regulations during COVID-19 across 44 European countries/regions [36].

Specifically, France has arranged for remote consultations (teleconsultations) [37] and allows for ‘the first intake of the drugs necessary to achieve a voluntary termination of pregnancy by medication (…) in the context of a teleconsultation with the doctor or midwife’ [38].

The United Kingdom government, in order to limit the transmission of the virus and guarantee access to abortion, has temporarily approved the use of abortion drugs at home up to 10 weeks’ gestation (9 weeks and 6 days) ‘following a telephone or e-consultation with a clinician, without the need to first attend a hospital or clinic’ [39].

A document issued by the RCOG entitled *Coronavirus (COVID-19) Infection and Abortion Care*, updated on 31 July 2020, reported the approval given by the Department Health and Social Care in England and the government of Wales for the use of mifepristone at home up until the tenth week of gestation. It added that ‘these new approvals permit medical practitioners to prescribe from home’. Further, it was reported that the government of Scotland had allowed the use of mifepristone and misoprostol at home without specifying a gestational period, thus deferring the decision as to the appropriacy of prescribing the procedure at home to healthcare personnel.

The same document stipulated that women in self-isolation who are suitable for medical abortion at home should be treated without the need for attendance in person at a clinic (e.g., treatment pack by post), ‘maximizing’ the use of remote consultations to provide pre- and post-abortion care [40].

In response to the spread of the SARS-CoV-2 virus, the Ministry of Health in Italy, by means of a circular dated 16 March 2020, provided ’general indications’ as regards rescheduling both outpatient and inpatient activities which, based on a risk/benefit analysis, were considered clinically deferrable [41]. In a clarification dated 30 March 2020, the Ministry specified certain activities that were to be considered non-deferrable, including VTPs [42]. Nevertheless, in April 2020, some spokespeople of the Parliament reported ‘serious difficulty in accessing voluntary terminations of pregnancy as an indirect consequence of the COVID-19 epidemiological emergency’. Ten years on from the approval of the ‘Guidelines for Voluntary Termination of Pregnancy by Mifepristone and Prostaglandin’, on 24 June 2010, the *Consiglio Superiore di Sanità* (Council of Health), in special session on 4 August 2020, expressed a favorable opinion regarding ‘recourse to voluntary termination of pregnancy by pharmacological method up to 63 days’ or 9 complete weeks’ gestation at properly equipped outpatient facilities/family counselling clinics, functionally connected to a hospital and authorized by the Region or a day hospital’. The Council also issued an opinion in favor of the new ‘Guidelines for Voluntary Termination of Pregnancy by Mifepristone and Prostaglandin’, which were attached. These guidelines set down the treatment admission criteria, which included ‘intrauterine pregnancy with amenorrhea/ultrasound date up to 63 days’ [43].

In line with AIFA Resolution No. 865/2020–Modification of Methods of Use of mifepristone-based medicine Mifegyne (RU486)–[44] dated 12 August 2020, mifepristone has been approved ‘in sequential association with a prostaglandin analogue up to the 63rd day (9 weeks) of gestation’. Article 3 of AIFA Resolution No. 1460/2009 regarding the restrictions of use was repealed (‘admission to one of the healthcare facilities listed in Art. 8 of Law 194/78 from ingestion of the medicine until the verification of the expulsion of the product of conception. The entire abortion path must take place under the supervision of a doctor of obstetric-gynecologic service who are asked to provide correct information on the use of the medicine (…) the drug must be taken within the seventh week of amenorrhea (…)’). The most recent modification also confirms the classification for the supply of Mifegyne as a medicinal product ‘subject to restricted medical prescription, for use in hospital or similar settings only, including the facilities listed in Art. 8 of Law No. 194, dated 22 May 1978′.

A comparison between previous regulations and current updates in Italy can be seen in Table 1, Table 2 and Table 3.

The updated version of the guidelines on medical abortion, which is inopportune despite its introduction during pandemic, gives rise to several interconnected points for consideration of a medico-legal nature. However, such innovations must be taken in context, not only with the general issues regarding the use of mifepristone, but also with issues regarding the medical management of voluntary termination of pregnancy. The extension of the permitted timeframe within which medical abortion is permitted calls into question the safety of the method, particularly given the health and mortality risks involved [45]. The emphasis given to the issue of related risks seems to be based on political and ethical rather than scientific leanings, since the Italian Society of Obstetrics and Gynecology (SIGO) has already expressed its opinion in that regard, as reported by the Council of Health [43], stating that ‘there is no scientific evidence that advise against the administration of these drugs between the seventh and ninth week’. The combination of the two medicines up to 63 days’ gestation (and sometimes beyond) has also been recognized and permitted by single states [15,18,39], scientific societies [19,35], and the WHO [20,21].

Prior to the publication of these updated guidelines, as part of the abovementioned opinion published by the Council of Health, the SIGO stated that based on the scientific literature, ‘Mifepristone may be administered, either in a family counselling or hospital outpatient clinic’ with possible subsequent home discharge, and that prostaglandin may be taken in hospital (the second time in day hospital) or alternatively at home, should the woman so desire. Furthermore, the SIGO highlighted that the USA, along with several European countries, had approved prostaglandin to be taken at home and that France had approved both medicines to be taken in non-hospital settings.

Recently, the International Federation of Gynecology and Obstetrics (FIGO), affirmed that ‘Telemedicine abortion programmes implemented during the pandemic have demonstrated that such services can provide efficacy, safety, efficiency, and acceptability without ultrasound scan’ and recommended investment to ‘strengthen the provision of and access to telemedicine’ [46].

AIFA Resolution No. 865/2020 repealed the article previously in force (No. 3, Resolution 1460/2009) pertaining to the usage constraints regarding mifepristone, confirming its classification as subject to ‘restricted medical prescription, for use in hospital or similar settings only, including the facilities listed in Art. 8 of Law No. 194, dated 22 May 1978′. Similarly, the Council of Health has published its opinion in favor of performing medical abortions in properly equipped outpatient/public family counselling clinics provided that they meet certain requirements or in day hospitals. In addition, the updated guidelines set forth that the procedure should be divided into specific stages: a pre-operation stage; outpatient attendance/day hospital admission including the administration of mifepristone and home discharge after 30 min (1st day); a home stage (2nd day); counselling clinic/outpatient/day hospital admission for the separated administration of 2–3 doses of prostaglandin and a transvaginal ultrasound to confirm completion of the abortion, as well as approximately two hours of further observation (3rd day); follow-up 14 days after taking prostaglandin.

The option of terminating pregnancies in outpatient facilities and family counselling clinics does represent a move forward as compared to the law previously in force, although it is not completely aligned with the provisions set down in other European states and in the USA. In fact, the procedure allows neither mifepristone nor misoprostol to be taken at home despite the evident advantages [20] and the widespread use of at least misoprostol at home on an international level.

Furthermore, the option to perform medical abortions in family counselling clinics creates various problems, especially of an organizational nature, as regards the need to allow healthcare professionals to exercise their freedom of conscience and at the same time ensure access to abortion medicines and pre- and post-procedure care at said clinics. Further issues may arise from the fact that, in such facilities, there is no ‘explicit provision to ensure the service’ [22]; Art. 9 of Law No. 194/78 [25], in reference to conscientious objection, sets down that ‘authorized hospitals and care homes are in any case required to ensure the performance of procedures’, with no mention of counselling clinics.

Medical termination of pregnancy is a choice for women, based on factors of a psychological and organizational nature, in line with the doctor’s recommendations as regards the appropriacy of and the absence of contraindications to the procedure. In accordance with Art. 8 of Law No. 194/78 [25], it is the doctor’s responsibility to assess potential contraindications to abortions and thus to identify clinical criteria (in compliance with the guidelines) and situations that may mean that a medical abortion is not advisable. The non-clinical criteria set down by the new guidelines to evaluate the exclusion of the woman from the procedure include the inability to reach an obstetrics–gynecology emergency room within one hour (whereas the previous guidelines used the general term ‘promptly’). Furthermore, the same guidelines advised that minors be offered regular admission, whereas prior to the recent update, medical abortion was considered ‘inadvisable’ (the previous guidelines also stated that ‘minors without parental consent should be excluded from this procedure’). Prescriptions for mifepristone remain the doctor’s responsibility, as defined in the 2020 AIFA Resolution.

Art. 3 of the 2009 AIFA Resolution previously in force deferred the task of providing patients with information regarding mifepristone and related risks to the doctor; this article was repealed by the 2020 Resolution. The updated guidelines make reference to an interview with a ‘healthcare operator’, thereby using a generic expression. In any case, in line with Art. 1(3) of Law No. 219, 22 December 2017 [47], the general responsibility to provide information on healthcare treatments prescribed for the woman, and in this specific case, to ensure the patient’s full comprehension of the procedure, lies with the doctor.

The new guidelines have made a further adjustment: whereas medical abortions could previously be performed as a regular admission only, now the procedure can be carried out in a day hospital and in outpatient/family counselling clinics. It follows that there is a choice of location in which to complete the medical abortion. The choice lies with the woman, notwithstanding the doctor’s responsibility to identify situations that require the procedure to be performed in a hospital setting (even if in a day hospital).

The responsibility therefore lies with doctors to recognize potential contraindications both to the medical abortion procedure and the administration of mifepristone and misoprostol in non-hospital settings.

## 4. Conclusions

In conclusion, the innovations brought in during the COVID-19 pandemic as regards terminations of pregnancy by mifepristone and misoprostol in Italy, both as regards the permitted timeframe and the location for the procedure, though not completely in line with provisions on an international level, represent a partial recognition of a woman’s need to undergo abortion procedures in the most comfortable setting possible. These innovations comply with the most recent exhortations of the European parliament to improve the existing abortion procedures, evaluate methods for handling any deficiencies that have come to light as a result of the pandemic, and ensure access to abortion ‘including by means of abortion pill’ [48].

These innovations do not affect doctors’ responsibilities, especially as pertaining to procedure-related indications and contraindications, the information provided to the woman, and to ensuring the woman’s consent to and comprehension of the procedure.

All these legal changes have provided women the opportunity to undergo the pharmacologically induced termination of pregnancy and often with less difficulty. However, said changes place new aspects of professional liability on clinicians, who are required to manage the request from patients,. Doctors, in particular gynecologists, are responsible for deciding whether the clinical situation of the woman is such to permit performance of a pharmacological prescription or to resort to hospitalization (even in day hospital), a choice which must be made on the basis of acceptable risks for the patient. Doctors are also responsible for the information they provide to the patient regarding the risks and possibilities of pharmacological abortion methods and, as a result, are required to provide appropriate healthcare services in line with the legal and clinical indications acquired and, if possible, in accord with the woman’s preferences, acquired by means of an interview and discussion.

## Figures and Tables

**Table 1 ijerph-18-12043-t001:** Comparison between previous and current opinion of Italian Council of Health on medical abortion.

*Consiglio Superiore di Sanità* (Council of Health), 18 March 2010 Session [30]	*Consiglio Superiore di Sanità* (Council of Health), 4 August 2020 Special Session [43]
(…) necessary for the purposes of ensuring compliance with Law No. 194/78 throughout the national territory, that the medical voluntary termination of pregnancy procedure be performed as a **regular admission until the verification of the complete expulsion of the product of conception**	Expressed a favorable opinion regarding recourse to voluntary terminations of pregnancy by pharmacological method up to **63 days’ or 9 complete weeks’ gestation** at a properly equipped **outpatient facilities/family** **counselling clinics**, connected to a hospital and authorized by the Region, or a **day hospital**

**Table 2 ijerph-18-12043-t002:** Comparison between previous and current guidelines on medical abortion in Italy.

Guidelines for Voluntary Termination of Pregnancy by Mifepristone and Prostaglandin Approved on 24 June 2010 [27]	Guidelines for Voluntary Termination of Pregnancy by Mifepristone and Prostaglandin, Attachment to Opinion Dated 4 August 2020 [43]
Treatment admission criteria based on: - Intrauterine pregnancy with amenorrhea up to **49 days/ultrasound date up to 35 days** - VTP request document/certificate - Completed and signed informed consent form - **Availability to regular admission until completion of the procedure** - Availability to complete follow-up check remotely within 14–21 days from discharge	Treatment admission criteria based on: - Intrauterine pregnancy with **amenorrhea/ultrasound date up to 63 days** - VTP request document/certificate - Completed and signed informed consent form - Availability to complete follow-up check remotely after 14 days from the administration of prostaglandin (misoprostol)

**Table 3 ijerph-18-12043-t003:** Comparison between previous and current Resolution of the Italian Medicines Agency.

Resolution No. 1460, 24 November 2009 [26]	Resolution No. 865/2020–Mifegyne (RU486), 12 August 2020 [44]
Therapeutic indications: medical termination of developing intrauterine pregnancy in sequential association with a prostaglandin analogue, which is authorized for use in Italy up to **49 days of amenorrhea** (…) Precautions for use: (…) **admission** into a healthcare facility listed in Art. 8 of Law No. 194/78 from medicine ingestion until the verification of the expulsion of the product of conception. The entire abortion path must take place under the supervision of a doctor of the obstetrics-gynecology service (…) Classification for supply: (…) for use in hospital or similar settings only, including the facilities listed in Art. 8 of Law No. 194, dated 22 May 1978	(…) authorizing the use of mifepristone (RU486)-based Mifegyne (…) in sequential association with a prostaglandin analog up to **63 days’ (equal to 9 weeks) of gestational age**. (…) Art. 3 of Resolution No. 1460 dated 24.11.2009 was repealed * Classification for supply: (…) classification for supply of Mifegyne (…) confirmed as follows: medicinal product subject to restricted medical prescription, for use in hospital or similar settings only, including the facilities listed in Art. 8 of Law No. 194, dated 22 May 1978 * Usage path constraints

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
