# Peer review of "Pharmacological Abortion in a Pandemic: An Italian Medico-Legal Perspective"

_ijerph, 2021, doi:10.3390/ijerph182212043_

Round 1

Reviewer 1 Report

Manuscript ID: ijerph-1424306

Title: Pharmacological abortion in a pandemic: a medico-legal perspective

Authors: Clara Cestonaro, Anna Aprile, Matteo Bolcato *, Daniele Rodriguez, Alessandro Feola, Giulio Di Mizio

Submitted to section: Women's Health

REVIEW

This manuscript was written by persons belonging to three institutions and different fields: 1. Legal Medicine, 2. Department of Experimental Medicine, and 3. Forensic Medicine. Such interprofessional and interdisciplinary approach is supposed to result in a very comprehensive paper, interesting for both medical and law students and physicians/practitioners. Unfortunately, the outcome is completely opposite to such expectations. In order to properly address the topic, authors need to involve public health expert and/or OBGYN Specialist.

Introduction

The first sentence in introduction: “An important aspect of public health, in terms of the organization of and access to healthcare services, is the protection of women’s health and autonomy by means of terminations of pregnancy” is written and commented by persons who do not belong to either public health or clinical work.

The manuscript does not compare the RESULTS/PARAMETERS/OUTCOMES of previous approach to medical termination of pregnancy before COVID pandemic with the existing ones during pandemic. Authors compared regulations dated on 18.03.2010 and 04.08.2020 presented on Tables 1-3 (rows 254-257).  Authors defined the “Aim” as follows: “The aim of this article is to compare the current regulations in force in Italy with those in other European countries as regards pharmacologically induced terminations of pregnancy, with particular reference to updates introduced in the context of the current pandemic”. In the aim stated it is not clear in which “other European countries”, assuming there are a lot of them, and why did they choose some countries, and skipped/excluded the others. Authors just mentioned some countries, but did not precisely assess their regulations on the matter.  Authors evaluated/mentioned the situation in the USA as well (rows 177-190), which is out of the scope defined in aims and written in the title.

Medical Abortion

Regarding the next section “Medical Abortion” I should write it is very strange for medical abortion and medications used for this purpose to be evaluated by persons being not clinicians and even belonging to law and forensic medicine, without the extensive information/deep knowledge about the entity (medical abortion) and medications used (mifepristone/misoprostol). If we accept such approach, that will mean anyone can write some sentences in the row (I am intentionally not using the formulation “Review”) belonging to any field in the society: sciences, medicine, law, agriculture, meteorology, etc. The Editors should not allow such approach, even almost a half of the Section’s editorial board comes from the same country as authors.

The reader will find the following sentences (rows 330-337): “The new Guidelines make a further adjustment: whereas medical abortions could previously be performed as a regular admission only, now the procedure can be carried out in a day hospital and in outpatient/family counselling clinics. It follows that there is a choice of location in which to complete the medical abortion. The choice lies with the woman, notwithstanding the doctor’s responsibility to identify situations that require the procedure to be performed in a hospital setting (even if in a day hospital). The responsibility therefore lies with doctors to recognize potential contraindications both to the medical abortion procedure and the administration of mifepristone and misoprostol in non-hospital settings”. This formulation is very good from the narrative point of view, and for the books in law, to put responsibilities on the “physician’s shoulders” – but, how to apply them in practical work exactly following the text written by the specialist in law. This makes the “hole”, or a “grey zone” for interpretation of precise responsibilities of doctors and their protection/support if sued. This is something which is out of the scope of this journal, does not belong to my expertise (I am not an expert in law but in the Clinical Medicine) and I don’t want to comment the last sentence in this paragraph, although being court expert in the field of OBGYN.

So, the text does not belong to either narrative review, or systematic review/meta-analysis. It is not a brief communication with the main leading point. It is supposed to be written as a comparison of legislative points in 2010 and 2020 in the field of medical abortion.  As the whole text is written in the way just explaining the medical abortion, and without experts in either the field of public health or OBGYN, the conclusions stayed in the same sense as the whole text.

In conclusion, it was written: “the innovations brought in during the Covid-19 pandemic as regards terminations of pregnancy by mifepristone and misoprostol in Italy, both as regards the permitted timeframe and the location for the procedure, though not completely in line with provisions on an international level, represent a partial recognition of a woman’s need to undergo abortion procedures in the most comfortable setting possible. These innovations comply with the most recent exhortations of the European parliament to improve the existing abortion procedures, evaluate methods for handling any deficiencies that have come to light as a result of the pandemic, and ensure access to abortions ‘including by means of abortion pill’. These innovations do not affect doctors’ responsibilities especially as pertain to procedure-related indications and contraindications, the location in which to perform medical abortions, the information to provide to the woman, and to ensuring the woman’s consent to and comprehension of the procedure.”

Considering the written conclusion, what will be the benefit for the person who is reading this text, if published - eventually. Assuming it has been submitted to the journal related to environmental and public health research, it is really not clear how the persons belonging to other professions and without any public health or clinician (or preferably: specialist in OBGYN) assisting or advising them for this topic, can discuss and explore such very sensitive entity.

In conclusion of this review, it is not clear:

  • What is a scientific and clinical contribution of the manuscript in the field of environmental and public health?
  • What is a ”take home message” to the reader?

Unfortunately, the authors invested the time and energy to write this text which is not useful for this journal. It can be explained by the fact that manuscript was written by persons belonging to: 1. Legal Medicine, 2. Department of Experimental Medicine, and 3. Forensic Medicine.

This text cannot be recomposed by the same authors, in order to be suitable for journal of environmental and public health. Therefore, it is not advised for further processing, and the only decision is: REJECTION.  

Best regards,

Author Response

We want to thank the reviewer for the time spent evaluating our work and for the comments. However, we would like to clarify that the authors are all medical examiners belonging to Italian academic institutions and that, even if they do not operate as gynecologists or doctors with clinical activity in the ward, they are engaged on a daily basis in the management of clinical forensic cases, i.e. in support of colleagues in resolving issues. inherent to patients that have medico-legal implications. We are committed to writing a research article for a specific special issue that deals with the legal changes on the complex issue of abortion and its medico-legal implications in light of the legal updates after the pandemic. We do not believe that in the absence of a gynecologist specialist we cannot write an article of this kind, on the contrary we think that our article written by medical examiners is of support and interest to clinical colleagues.

By reading the article, the reader will be able to get a complete idea of ​​the changes regarding the drugs used, the reasons that prompted these changes and get a real update on European legislation in this matter. It is not only an end in itself knowledge but also applicable to clinical situations and can allow further ethical and deontological reflections. We therefore believe that the article deserves interest and publication. Thanks for the attention

Reviewer 2 Report

I would like to congratulate the authors for preparing this nice perspective. I enjoyed reading it.

I have one question: What is the impact of the use of teleconsultation in the UK (or other countries) on the abortion outcome? Are there any data about this issue?

Reviewer 3 Report

This article provides a summary of the regulations surrounding pharmacological abortion in Europe and the US with a special focus on Italy.  While an important topic, I had the following questions and suggestions:

1) It is unclear from the title and the abstract that the focus will be so strongly on Italy.

2)  The authors summarized some of the COVID response to abortion from the US and Europe. It would be helpful if the authors did a more formal comparison with Italy.

3)  Can the authors provide more context why and how Italy's response is similar/ differ from Europe?

4)  Can the authors provide a current update of the regulations as many countries have begun to lift Covid restrictions?

Round 2

Reviewer 1 Report

Dear Sirs,

The revised manuscript did not clarify previously raised questions:

  • What is a scientific and clinical contribution of the manuscript in the field of environmental and public health?
  • What is a ”take home message” to the reader?

Unfortunately, this manuscript it is not advised for further processing, and the only decision is: REJECTION.  

Please do not send me again for further revision.

Best wishes,

Author Response

We want to thank the reviewer again for his time. We have again tried to meet the suggestions made. We hope for a positive reception.

Yours sincerely

Reviewer 3 Report

It is unclear to me what is the scientific contribution of this paper.  It provides a selected summary of pharmacological abortion during Covid with a focus on changes in Italy. However, the authors aren't clear about the purpose of this paper beyond providing this summary.

Author Response

We thank the reviewer again for his suggestion. This article summarizes regulations concerning pharmacological abortion in force prior to pandemic and after Sars-CoV-2 virus spread, particularly focusing on Italian updates. Previous and current guidelines were explained and compared, highlighting doctors' responsibilities in light of recent modifications. The article is therefore important in order to clarify to clinicians the responsibilities that still lies with them despite guidelines' updates. To try to enhance this message, we have modified the conclusions in a more analytical way. We believe that this article may be of interest to clinical colleagues who are faced with having to use these drugs and to health authorities who have to take note of these legal innovations.